# The spatiotemporal evolution and driving factors of urban–rural integration over the past eight years in China: Evidence from 31 provinces

Hui Liu, Wei Wang *

College of Public Administration and Law, Hunan Agricultural University, Changsha, China

* weiw@hunau.edu.cn

## Abstract

Urban-rural integration development is essential for overcoming the inherent barriers between urban and rural areas and achieving coordinated regional development. This study uses panel data, along with methods such as the entropy method, coupling coordination degree model, and geographic detector model, to analyze the spatiotemporal evolution and main influencing factors of urban-rural integration development across 31 provinces, autonomous regions, and municipalities in China from 2015 to 2022. The findings indicate that: (1) The overall level of urban-rural integration in China shows an upward trend, but the growth rate is uneven, exhibiting a phased pattern of "rapid growth—slow development—fluctuating rise." (2) There are significant regional differences, with eastern regions leading in development, central regions rising rapidly, and western regions showing huge potential. (3) Key driving factors promoting urban-rural integration include internet broadband access, per capita disposable income, education expenditure, unemployment insurance coverage, and greening coverage. Based on these findings, it is recommended to develop phased strategies, implement a regional gradient development approach, and prioritize strengthening key areas to systematically promote urban-rural integration.

## Introduction

The relationship between urban and rural areas, as a fundamental relationship in human social development, has always been a significant theoretical and practical issue concerning the national economy and people's livelihoods [1]. In the historical development of China, the relationship between urban and rural areas has undergone several transformations, which can be roughly summarized into three stages: urban-rural chaos, urban-rural opposition, and urban-rural integration [2]. In the early stages of primitive society, due to the lack of technological means, the production methods were primitive and simple, highly dependent on nature's gifts. As a result, the boundary between cities and rural areas was not clearly defined, and there was

**Data availability statement:** All relevant data are within the manuscript and its Supporting information files.

**Funding:** This research was funded by the National Social Science Late Grant Program, grant number 22FGLB007 to W.W.; Project of Natural Science Foundation of Hunan Province, grant number 2023JJ30313 to W.W.; Hunan Provincial Graduate Research Innovation Project, grant number CX20251057 to H.L.; Graduate Research Innovation Project of Hunan Agricultural University, grant number 2025xkc124 to H.L.

**Competing interests:** The authors have declared that no competing interests exist.

no distinct urban-rural difference [3]. However, with the rise and development of capitalism, cities quickly became the core of development, attracting a large influx of rural populations. Farmers gradually became wage workers within the capitalist production system, and a clear divide emerged between urban and rural areas in economic, social, and cultural aspects [4,5]. Fortunately, continuous technological advancements brought new development opportunities to rural areas [6]. The improvement in agricultural productivity freed up a large labor force in rural areas, which was transferred to more value-added industries, such as agricultural product processing and rural tourism, bringing higher income levels to farmers [7]. Meanwhile, the optimization and upgrading of the industrial structure not only enhanced the competitiveness of traditional agriculture but also gave birth to emerging sectors like modern agriculture and ecological agriculture, promoting the diversification of rural economies [8,9]. Against this backdrop, urban-rural integration development gradually gained attention and became an important strategic direction for national development. In 2019, the Central Committee of the Communist Party of China and the State Council issued the "Opinions on Establishing and Improving the System and Policy Framework for Urban-Rural Integration Development," which clearly pointed out the need to accelerate the formation of a system mechanism for the smooth flow of urban and rural elements and the balanced allocation of public resources, to build a new type of urban-rural relationship that promotes mutual advancement between industry and agriculture, complements urban and rural areas, ensures coordinated development, and fosters common prosperity [10].

Although urban-rural integration has become an important strategic direction for national development, most studies, constrained by the traditional urban-rural dual analysis framework, have failed to reveal the intrinsic mechanisms of urban-rural integration from a holistic perspective, limiting a deeper understanding of the dynamic relationship between urban and rural areas. At the same time, existing literature tends to focus on traditional dimensions such as the economy, society, and environment, with insufficient attention given to key dimensions such as infrastructure connectivity and social security. This has made it difficult to fully explain the multi-dimensional driving pathways of urban-rural integration. To address these issues, this study adopts an urban-rural integration perspective and constructs a five-dimensional integrated analysis framework, including "infrastructure—economic development—social services—social security—ecological environment," filling the gap in existing research. This framework not only contributes theoretically to deepening the understanding of the multi-dimensional collaborative mechanisms of urban-rural integration, but also provides scientific, reasonable, and efficient empirical evidence for formulating differentiated urban-rural integration policies at the practical level.

## Literature review

The concept of urban–rural integration derives from Thomas More's seminal work, Utopia [1]. In that work, More described an ideal society in which urban and rural regions were cohesively integrated, operating as a singular entity. In his concept, resource allocation, population distribution, and production activities were governed

by need-based distribution, fostering societal resources' equitable allocation and effective exploitation [11]. This concept was as an intellectual precursor to subsequent urban–rural integration theories. At the end of the 19th century, Engels provided a more comprehensive theoretical framework in The Principles of Communism [12]. He asserted that, to achieve urban–rural integration, antiquated labor divisions needed to be eliminated. He proposed that this could be achieved through cohesive production and education frameworks, facilitating adaptable employment transitions and an equal distribution of benefits, allowing all citizens to collaboratively generate and reap the rewards of shared prosperity [2]. Engels also contended that such measures would promote the holistic development of human skills while enhancing urban–rural integration. A prominent early urban planner, Ebenezer Howard contributed to this discussion in Garden Cities of Tomorrow. By integrating urban–rural principles into his urban planning methodologies, Howard converted these concepts into implementable models, grounding the idea in actual applications [13]. In sum, early urban–rural integration research, mainly influenced by Western utopian socialism, includes two fundamental perspectives: "urban-led rural development" and "rural-promoted urban development." These perspectives provide a significant theoretical starting point for enhancing modern urban–rural integration strategies. On this basis, further theoretical doctrines have been developed, including the "Urban Bias Theory" "Rural Bias Theory" "Urban-Rural Dual Structure Theory" and "Urban-Rural Integrated Development Theory". The Urban Bias Theory advocates for an urban-centered development path, emphasizing that policies and resources should concentrate in urban areas [14]. However, this often leads to the stagnation of rural development and a widening development gap between urban and rural areas. The Rural Bias Theory, in contrast, focuses on the intrinsic dynamics of rural areas, advocating for sustainable progress based on rural resources [15]. However, this theory somewhat neglects the role of urban areas in driving and influencing rural development, and in practice, it often falls into the trap of isolated development. The Urban-Rural Dual Structure Theory focuses on the structural contradictions caused by the coexistence of traditional and modern sectors, revealing how institutional divisions constrain resource flow and social equity [16]. However, its analysis tends to be static, lacking sufficient explanation of the dynamic mechanisms of urban-rural interaction and integration [17]. The Urban-Rural Integrated Development Theory, on the other hand, aims to break down the development barriers between urban and rural areas, advocating for complementary functions, institutional coordination, and systemic integration, promoting two-way flows of elements and resource sharing, and providing a theoretical direction for achieving genuine integrated development [1].As a multidimensional concept, urban–rural integration revolves around two entities: the "urban" and the "rural." Urban areas, as unique geographical systems, are defined by aggregating non-agricultural enterprises and residents, whereas rural areas include all regions outside the urban limits [18]. From a geographical standpoint, no distinct territorial boundary separates urban and rural regions. Their distinctions are predominantly evident in their functions rather than in their geographical arrangement. Cities deliver nonagricultural products, services, and related employment possibilities, whereas rural areas primarily focus on agricultural production, supplying agricultural products and services to society [19]. This functional mismatch has led to issues such as obstructed resource flow, slow industrial development, disproportionate public services and severe rural environmental pollution, which have greatly hindered urban-rural integration development. To address the aforementioned issues and promote urban-rural integration development, scholars have proposed a series of comprehensive measures, including promoting industrial coordinated development to facilitate complementary and mutually beneficial urban-rural economies [20]; striving to achieve the equalization of public services to ensure that urban and rural residents enjoy equal social welfare [21]; and emphasizing the strengthening of ecological environmental protection to ensure the sustainability of urban-rural development [22]. These comprehensive measures form an all-encompassing strategy for promoting urban-rural integration development.

In summary, existing research on urban-rural integration mainly focuses on theoretical foundations, core concepts, practical exploration, and policy recommendations. These achievements are largely based on normative analytical frameworks, providing important references for the field. However, constrained by traditional research perspectives and analytical frameworks, existing studies still have notable shortcomings in systematically understanding the mechanisms

of urban-rural interaction and the integration of multi-dimensional elements. To fill this research gap, this study adopts a holistic perspective on urban-rural integration and constructs a five-dimensional analysis framework that includes "infrastructure—economic development—social services—social security—ecological environment." Within this framework, infrastructure is regarded as the material foundation for the flow of urban and rural elements, while social security serves as an important institutional support for promoting social equity. Together, these dimensions provide a new analytical perspective for understanding urban-rural integration. Based on panel data from 31 provinces in China from 2015 to 2022, this study employs methods such as the entropy weight method, coupling coordination degree model, and geographic detector to systematically examine the national level of urban-rural integration, its temporal and spatial evolution characteristics, and key driving mechanisms. The aim is to provide more comprehensive theoretical support and empirical references for achieving coordinated urban-rural development. Fig 1 illustrates the specific steps of our research.

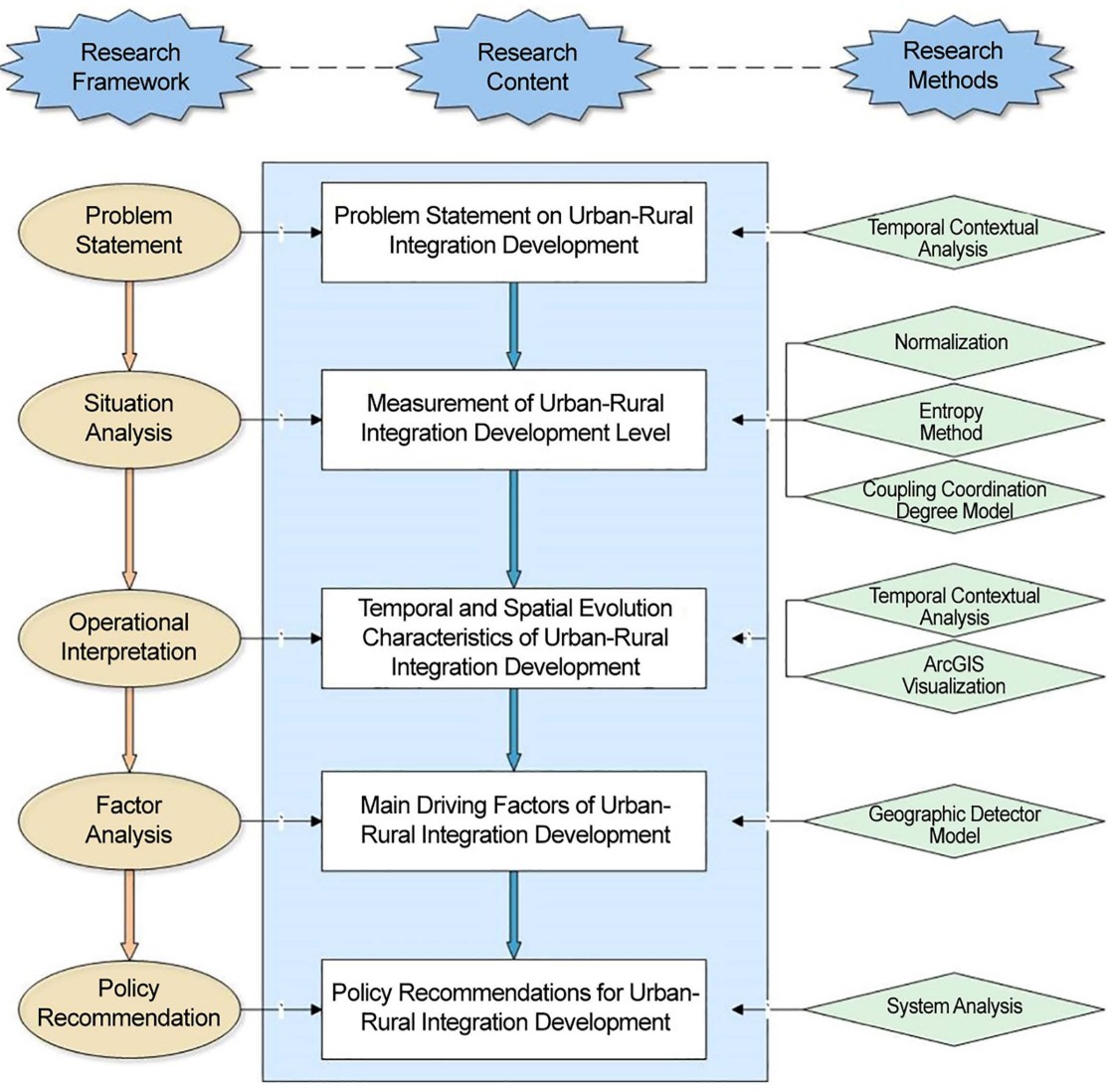

**Fig 1. Research structure.**

## Methods

### Study area

This study focuses on the 31 provinces, autonomous regions, and municipalities directly under the central government of China, specifically covering Beijing, Tianjin, Hebei, Shanxi, Inner Mongolia Autonomous Region, Liaoning, Jilin, Heilongjiang, Shanghai, Jiangsu, Zhejiang, Anhui, Fujian, Jiangxi, Shandong, Henan, Hubei, Hunan, Guangdong, Guangxi Zhuang Autonomous Region, Hainan, Chongqing, Sichuan, Guizhou, Yunnan, Tibet Autonomous Region, Shaanxi, Gansu, Qinghai, Ningxia Hui Autonomous Region, and Xinjiang Uygur Autonomous Region. It should be noted that, due to the unavailability of relevant data for Hong Kong Special Administrative Region, Macau Special Administrative Region, and Taiwan Province, these regions are not included in this study to ensure the scientific rigor and data accuracy of the research. The specific details are shown in Fig 2.

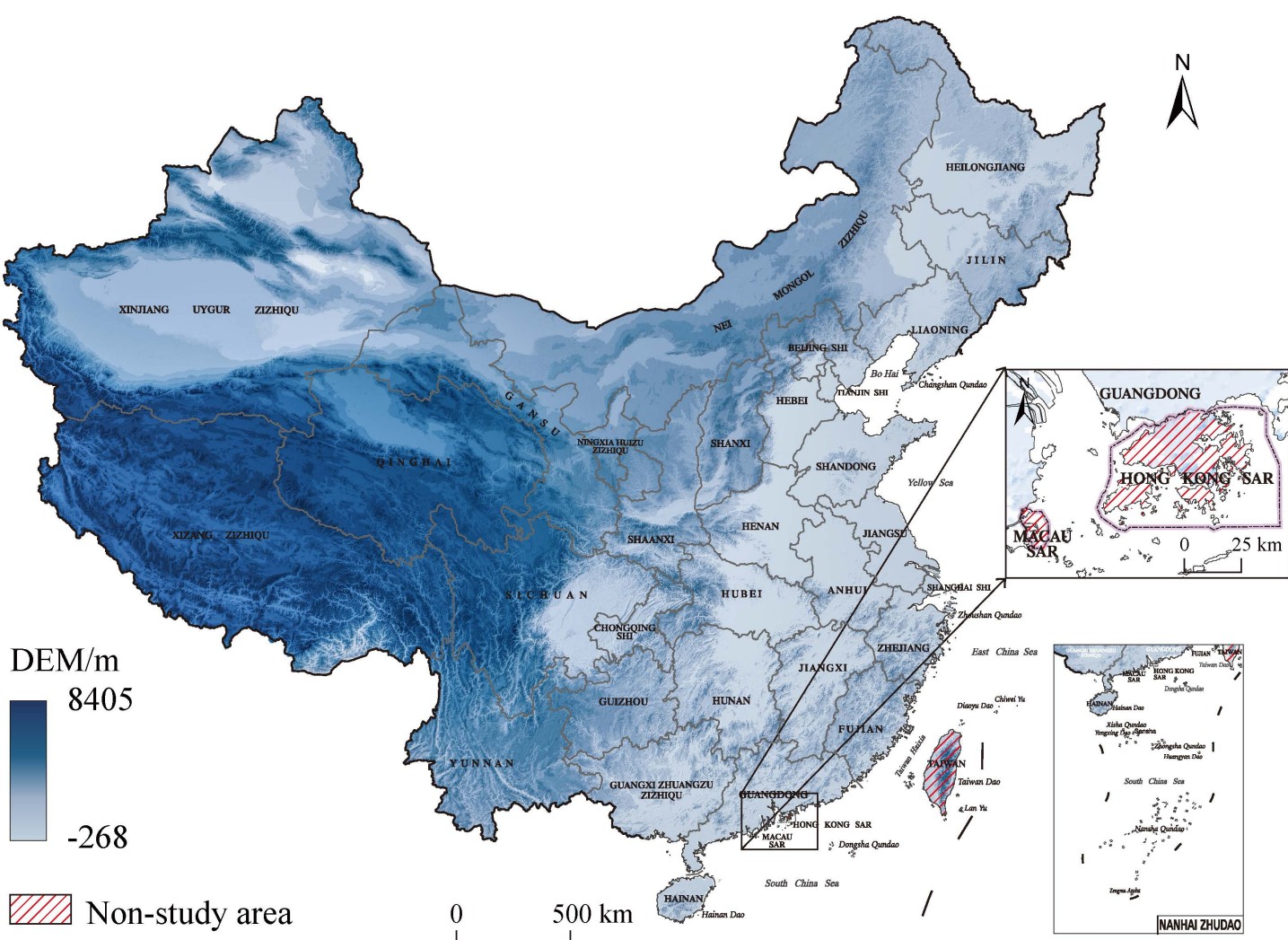

**Fig 2. Research area overview map.** (The base map of this figure uses public domain data from Natural Earth (available at https://www.naturalearth-data.com/), and the map was created using ArcGIS).

## Data sources

The data used in this study all come from official and authoritative channels, primarily including macroeconomic statistical data released by the National Bureau of Statistics and the Ministry of Industry and Information Technology, as well as national-level comprehensive statistical yearbooks such as the China Statistical Yearbook and the China Urban and Rural Construction Statistical Yearbook. Additionally, we have integrated relevant data from provincial, autonomous region, and municipal statistical yearbooks to ensure comprehensive coverage of urban-rural development conditions across different regions.

Regarding the data structure, this study constructs a panel dataset covering 31 provinces, autonomous regions, and municipalities in China, with a time span of 8 years, resulting in a total of 7936 valid units. The variables are mainly of two types: one is the comparative variable (e.g., urban-rural road length comparison coefficient), used to directly measure the development gap between urban and rural areas; the other is the descriptive variable (e.g., urban-rural pension insurance coverage rate), used to characterize the development level of specific dimensions.

It is worth noting that during the data processing, a small amount of missing data was inevitably encountered. To address this issue, linear interpolation was employed to reasonably fill the gaps, ensuring data completeness and the continuity of the analysis. Compared to other methods, linear interpolation offers significant advantages when dealing with small amounts of missing data and when the data generally exhibits a linear trend. It provides more precise approximations and effectively preserves the original characteristics of the data. However, we must also point out that this method is based on the assumption of linear changes in the data, which may introduce bias when there are large fluctuations in the variables or non-linear trends, thereby causing uncertainty in the analysis of individual variables. It should be noted that, due to the extremely low proportion of missing values in this study (less than 1%), and after comparing the statistical characteristics of key indicators before and after interpolation, no significant differences were found. Therefore, it can be considered that the bias introduced by linear interpolation is within a controllable range and will not have a significant impact on the accuracy of the overall research conclusions.

## Indicator descriptions

To scientifically measure the level of urban-rural integration, this paper constructs a comprehensive evaluation index system covering five major subsystems: infrastructure, economic development, public services, social security, and ecological environment. The core idea of urban-rural integration is to gradually reduce the gap between urban and rural areas and achieve coordinated development. Based on this goal, this study uses both comparative and descriptive indicators to represent the level of integration. Comparative indicators (such as the "urban-rural road length comparison coefficient") have 1 as the ideal balanced state, and the degree to which their values deviate from 1 reflects the size of the urban-rural gap. These indicators are typically expressed as the urban value divided by the rural value. The larger the value, the greater the urban-rural gap and the lower the degree of integration, which is negatively related to the integration goal, thus marked with a "−" attribute. Descriptive indicators (such as the "urban-rural pension insurance coverage rate") directly represent the development level of a particular dimension. The higher the value, the more significant the progress in that area, and the more positive the effect on urban-rural integration. Therefore, these are assigned a "+" attribute. The specific content is as follows.

Infrastructure constitutes the material foundation of urban–rural integration, supporting advancement and interconnectivity across geographies [23]. Advanced physical infrastructure increases transit networks between urban and rural areas, facilitates information interchange, and improves the overall quality of life in both settings [24]. To assess infrastructure development across geographical areas, we focused on several comparative coefficients. We used the comparative coefficient of urban-to-rural road length as the crucial indicator of the differences in transportation network configuration and road construction to assess the extent of physical infrastructure development [25]. We used the comparative coefficient

of broadband Internet access to reflect inequalities in information network infrastructure, representing a vital metric for urban–rural informatization [26,27]; and we used the comparative coefficient of water supply coverage between urban and rural areas to reflect disparities in water infrastructure, which directly affect living conditions in these locations [28].

Economic development is a growth catalyst for urban–rural integration. Efficient economic growth enhances per capita income, improves industrial structures, and raises consumption levels in both urban and rural areas [29,30]. To assess economic development across the provinces, we focused on three comparative coefficients. We used the comparative coefficient of per capita disposable income to demonstrate income differences and as a metric to assess the (dis)equilibrium of urban–rural economic development [31]. We used the urban–rural dual comparison coefficient to assess the ratio of added value and employment in the primary sector to the secondary and tertiary industries to underscore disparities in industrial structures and facilitate the analysis of economic factors [32,33]; and we used the comparative coefficient of total retail sales of consumer products to identify consumption differences between the urban and rural populations and provide insight into the integration of urban and rural consumer markets [1].

Public services are essential for urban–rural integration. Public services need to be equitable, meet the education requirements of both urban and rural populations, elevate cultural standards, and improve transportation and communication networks [21]. To assess public services across urban and rural areas, we again focused on three comparative coefficients. We used the comparative coefficient of per capita education expenditure between urban and rural populations to identify discrepancies in access to educational resources and as a crucial statistic for assessing equity in urban–rural schooling [34]. We used the comparative coefficient of per capita cultural and recreational expenditure to demonstrate disparities in cultural consumption levels and to provide insight into the integration of urban and rural cultural development [35]; and we used the comparative coefficient of per capita transportation and communication expenditure to highlight connectivity and information-sharing inequalities and to assess the interaction between urban and rural areas [36].

Social security serves as a support system for both urban and rural populations, and thus is important for their integration. An effective social security system can mitigate issues associated with age-related care, health hazards, and unemployment, providing an equitable basis for urban–rural integration [37,38]. To assess social security system development in the provinces, we focused again on three comparative coefficients. We used the comparative coefficient of pension insurance coverage to assess differences in older adult life security, functioning as a significant indicator of equity in urban–rural social welfare [34,39]. We used the comparative coefficient of medical insurance coverage to capture disparities in health risks and assess access to medical services to evaluate the equity in urban–rural healthcare systems [40]; and we used the comparative coefficient of unemployment insurance coverage to reflect differences in risk mitigation and reemployment assistance to assess employment security equity across urban and rural regions [41,42].

Ecological environment is fundamental for sustained urban–rural integration. An exemplary ecological environment improves citizens' quality of life and fosters green, sustainable development [30,43]. To assess the ecological environments across the rural and urban areas, we focused on three comparative coefficients. We used the comparative coefficient of greening coverage to identify differences in green infrastructure, an essential metric of ecological integration [25,44]. We used the comparative coefficient of sewage treatment rates to reflect environmental protection capabilities, offering insight into the degree of ecological preservation in urban and rural regions [45]; and we used the comparative coefficient of household waste treatment rates to highlight the inequalities in domestic waste management and as a crucial tool for evaluating environmental governance [46]. The details of these indicators are listed in Table 1.

## Modeling

**Entropy weighting.** The entropy weighting method is an objective weighting technique commonly employed in complex assessments [47]. This method computes the entropy of the indicators to evaluate the quantity of information they provide, establishing their corresponding weights in the assessment process. Indicators exhibiting elevated entropy signify increased system disorder and convey less information, resulting in reduced weights, whereas indications

 

**Table 1. Evaluation indicator framework measuring urban–rural integration in China.**

| Primary Indicator | Secondary Indicator | Indicator Name | Indicator Description | Attribute | Weight |
|---|---|---|---|---|---|
| Urban–Rural Integration | Infrastructure | Urban–rural road length comparison coefficient | Urban road length/ Rural road length/ % | − | 10.67% |
| | | Urban–rural broadband Internet access comparison coefficient | Urban broadband Internet access users/ Rural broadband Internet access users/ % | − | 3.47% |
| | | Urban–rural water supply coverage comparison coefficient | Urban water supply coverage/ Rural water supply coverage/ % | − | 4.98% |
| | Economic Development | Urban–rural per capita disposable income comparison coefficient | Per capita disposable income of urban residents/ Per capita disposable income of rural people/ % | − | 10.22% |
| | | Urban–rural dual comparison coefficient | (Value generated by primary industry/ Number of primary industry workers)/ (Value added by secondary and tertiary industries/ Number of secondary and tertiary industry workers)/ % | − | 6.10% |
| | | Urban–rural total retail sales of consumer goods comparison coefficient | Total retail sales of consumer goods in urban areas/Total retail sales of consumer goods in rural areas/ % | − | 6.65% |
| | Public Services | Urban–rural per capita education expenditure comparison coefficient | Per capita education expenditure of urban residents/ Per capita education expenditure of rural people/ % | − | 3.36% |
| | | Urban–rural per capita cultural and recreational expenditure comparison coefficient | Per capita cultural and recreational spending of urban residents/ Per capita cultural and recreational expenditure of rural people/% | − | 10.26% |
| | | Urban–rural per capita transportation and communication expenditure comparison coefficient | Per capita transportation and communication expenditure of urban people / per capita transportation and communication expenditure of rural residents/ % | − | 5.45% |
| | Social Security | Urban–rural pension insurance coverage comparison coefficient | Coverage of basic pension insurance for urban and rural residents/ Permanent population/ % | + | 4.66% |
| | | Urban–rural medical insurance coverage comparison coefficient | Primary medical insurance coverage for urban and rural residents/ Permanent population/ % | + | 6.86% |
| | | Urban–rural unemployment insurance coverage comparison coefficient | Unemployment insurance coverage for urban and rural residents/ Permanent population/% | + | 4.33% |
| | Ecological Environment | Urban–rural greening coverage comparison coefficient | Urban greening coverage ratio/ Rural greening coverage ratio/ % | − | 8.70% |
| | | Urban–rural sewage treatment rate comparison coefficient | Urban sewage treatment percentage/ Rural sewage treatment percentage/ % | − | 8.20% |
| | | Urban–rural household waste treatment rate comparison coefficient | Urban home waste management rate/ Rural household waste management rate/ % | − | 6.08% |

with lower entropy denote enhanced system order and better information, resulting in higher weights. Compared with conventional expert scoring techniques, the entropy method eliminates the subjectivity associated with manual scoring, thereby enhancing the reliability and objectivity of the data analysis. The precise steps are as follows:

Determine the ratio of indicators $P_{ij}$:

$$P_{ij} = \frac{X'_{ij}}{\sum_{i=1}^{n} X'_{ij}}$$

(1)

Compute the entropy of the indicators $E_j$:

$$E_j = -K \sum_{i=1}^{n} P_{ij} \ln P_{ij}$$

(2)

Compute the information utility value $D_j$:

$$D_j = 1 - E_j \tag{3}$$

Determine the weights $W_j$:

$$W_j = 1 - \frac{E_j}{\sum_{j=1}^{m} D_j} \tag{4}$$

where $i$ denotes the $i^{th}$ year; $j$ represents the $j^{th}$ indicator; $k = 1/\ln^n$, where n signifies the number of years; $0 \leq E_j \leq 1$; if $P_{ij} = 0$, then assign $P_{ij}\ln P_{ij} = 0$.

**Coupling coordination degree.** The coupling coordination degree model is a quantitative analytical instrument used to evaluate the extent of mutual effects and interactions among two or more systems [48]. The model includes a coupling coordination function that amalgamates the individual developmental level of each system and their interconnections, thus ascertaining the overall state of coupling and coordination among the systems. In contrast to other analytical methods, the coupling coordination degree model measures the intensity of interconnections across systems and reveals the coordination dynamics among the different elements in each system. The precise equations are as follows:

$$C = 5 \times \left\{ \frac{U_1 \times U_2 \times U_3 \times U_4 \times U_5}{(U_1 + U_2 + U_3 + U_4 U_5)^2} \right\}^{\frac{1}{5}} \tag{5}$$

$$T = \beta_1 U_1 + \beta_2 U_2 + \beta_3 U_3 + \beta_4 U_4 + \beta_5 U_5 \tag{6}$$

$$D = \sqrt{C \times T} \tag{7}$$

where C denotes the coupling degree; T is the comprehensive development level; and D is the coupling coordination degree. $U_1$, $U_2$, $U_3$, $U_4$ and $U_5$ denote the composite index scores for infrastructure, economic development, public services, social security, and the ecological environments, respectively. $\beta_1$, $\beta_2$, $\beta_3$, $\beta_4$ and $\beta_5$ are the coefficients of interest. We assume that the five subsystems hold equal significance; hence, we allocate a value of 1/5 for each. By establishing classification criteria in coordination degree research, the coordination degree can be categorized into 10 classifications, as shown in Table 2.

**Geographic detector model.** The geographic detector is a statistical instrument that identifies the principal elements affecting geographic phenomena. Based on spatial heterogeneity, it measures the intensity of spatial correlations between variables and geographic phenomena, thereby assessing the impact of each variable on the spatial patterns [49]. In

**Table 2. Criteria for classifying the degree of coupling coordination.**

| Coupling Coordination Degree D Value | Coordination Type | Coupling Coordination Degree D Value | Coordination Type |
|---|---|---|---|
| 0.0~0.1 | Extreme Discrepancy | 0.5~0.6 | Slight Coordination |
| 0.1~0.2 | Severe Discrepancy | 0.6~0.7 | Mild Coordination |
| 0.2~0.3 | Moderate Discrepancy | 0.7~0.8 | Moderate Coordination |
| 0.3~0.4 | Mild Discrepancy | 0.8~0.9 | High Coordination |
| 0.4~0.5 | Slight Discrepancy | 0.9~1.0 | Extreme Coordination |

contrast to conventional techniques, the geographic detector model evaluates the individual impact of each component and reveals the interactions among many factors, thereby providing a more thorough understanding of the underlying mechanisms. The precise equation is as follows:

$$q = 1 - \frac{1}{N\sigma^2} \sum_{h=1}^{L} N_h \sigma_h^2$$

(8)

where q denotes the value of the detector factor; $h = 1$; L denotes the stratification of variable X; N and $N_h$ signify the sample sizes of the study region and detection area, respectively; and $\sigma^2$ and $\sigma_h^2$ indicate the variance of variable Y in the study area and detection area, respectively. The q value ranges from 0 to 1, where a better q value signifies a more robust explanatory influence of factor X on variable Y.

## Results analysis

Our analysis examined comprehensive data to assess the progress of urban–rural integration levels in various regions of China, carefully considering the coordination degree values from 2015 to 2022. Our results offer insight into the efficacy of regional initiatives aimed at promoting urban–rural integration and establishes a solid basis for guiding future policy decisions. We summarize the integration degree by year in Table 3.

### Temporal evolution of urban–rural integration development in China

We found that the trajectory of urban–rural integration in China showed an upward tendency, although with unequal growth rates, typified by a pattern with phases of rapid growth, slow development, and a fluctuating rise. During the rapid expansion phase (2015–2018), the urban–rural integration coupling coordination degree increased markedly from 0.2931 to 0.5510, reflecting an average growth rate of 22.64%. The most significant advancement occurred from 0.4092 to 0.5510, indicating an impressive increase of 34.65%, the highest singular growth in this period. This signified a pivotal phase of rapid urban–rural convergence. In contrast, the gradual development period (2018–2019) experienced a minimal increase in the coupling coordination degree, from 0.5510 to 0.5520, with an average growth rate of merely 0.18%. This phase showed minimal growth throughout all intervals, indicating a developmental bottleneck in which integration slowed markedly. The fluctuating phase (2019–2022) reflected new acceleration in urban–rural integration, with the coupling coordination degree increasing from 0.5520 to 0.8141, at an average growth rate of 13.99%. This phase was also volatile. The initial interval, ranging from 0.5520 to 0.6746, saw substantial growth of 22.21%, followed by a decrease from 0.6746 to 0.7227, where the growth fell to 7.13%, suggesting growth fatigue. However, the concluding interval, ranging from 0.7227 to 0.8141, showed a reacceleration in growth to 12.65%. Overall, this phase represented a general upward trend despite volatility and inconsistent progress. Fig 3 illustrates these trends.

### Spatial distribution of urban–rural integration development in China

We found several spatial differences in China's urban–rural integration development, resulting in a varied pattern in which the eastern coastal regions dominated rural–urban integration, the central regions showed swift advancement, and the western regions demonstrate considerable potential. Our analysis reflected an important geographical proximity effect, wherein adjacent regions were able to exert reciprocal influence and support for each other, encouraging a tendency for regional interconnectedness development.

 In 2015, the eastern coastal areas spearheaded the urban–rural integration growth owing to their strategic geographic position, solid economic base, and openness. Tianjin (0.625), Guangdong (0.559), and Zhejiang (0.455) exhibited urban–rural integration indices highly exceeding the national average, establishing them as national benchmarks. By 2019, the

**Table 3. Index of urban–rural integration development across China regions from 2015 to 2020.**

| Region | 2015 | 2016 | 2017 | 2018 | 2019 | 2020 | 2021 | 2022 |
|---|---|---|---|---|---|---|---|---|
| Beijing | 0.343 | 0.414 | 0.528 | 0.893 | 0.825 | 0.439 | 0.49 | 0.429 |
| Tianjin | 0.138 | 0.336 | 0.588 | 0.824 | 0.847 | 0.872 | 0.935 | 0.941 |
| Hebei | 0.348 | 0.341 | 0.427 | 0.438 | 0.416 | 0.627 | 0.367 | 0.525 |
| Shanxi | 0.518 | 0.631 | 0.376 | 0.4 | 0.671 | 0.48 | 0.474 | 0.408 |
| Inner Mongolia | 0.441 | 0.396 | 0.404 | 0.485 | 0.588 | 0.391 | 0.419 | 0.265 |
| Liaoning | 0.417 | 0.346 | 0.34 | 0.553 | 0.796 | 0.467 | 0.436 | 0.404 |
| Jilin | 0.336 | 0.878 | 0.363 | 0.478 | 0.639 | 0.495 | 0.437 | 0.225 |
| Heilongjiang | 0.599 | 0.752 | 0.684 | 0.604 | 0.488 | 0.601 | 0.466 | 0.25 |
| Shanghai | 0.239 | 0.631 | 0.656 | 0.557 | 0.696 | 0.759 | 0.698 | 0.397 |
| Jiangsu | 0.278 | 0.34 | 0.342 | 0.491 | 0.659 | 0.672 | 0.625 | 0.397 |
| Zhejiang | 0.535 | 0.73 | 0.735 | 0.561 | 0.628 | 0.458 | 0.617 | 0.221 |
| Anhui | 0.517 | 0.523 | 0.509 | 0.409 | 0.555 | 0.286 | 0.643 | 0.602 |
| Fujian | 0.388 | 0.584 | 0.745 | 0.507 | 0.56 | 0.327 | 0.361 | 0.422 |
| Jiangxi | 0.455 | 0.38 | 0.405 | 0.671 | 0.566 | 0.324 | 0.539 | 0.368 |
| Shandong | 0.379 | 0.249 | 0.418 | 0.444 | 0.453 | 0.551 | 0.779 | 0.508 |
| Henan | 0.305 | 0.264 | 0.589 | 0.56 | 0.49 | 0.469 | 0.689 | 0.432 |
| Hubei | 0.415 | 0.553 | 0.544 | 0.374 | 0.35 | 0.34 | 0.42 | 0.422 |
| Hunan | 0.48 | 0.441 | 0.754 | 0.451 | 0.341 | 0.556 | 0.555 | 0.383 |
| Guangdong | 0.402 | 0.273 | 0.523 | 0.614 | 0.716 | 0.549 | 0.489 | 0.375 |
| Guangxi | 0.543 | 0.611 | 0.584 | 0.437 | 0.563 | 0.463 | 0.299 | 0.159 |
| Hainan | 0.514 | 0.686 | 0.512 | 0.809 | 0.832 | 0.419 | 0.409 | 0.275 |
| Chongqing | 0.841 | 0.637 | 0.438 | 0.423 | 0.402 | 0.337 | 0.294 | 0.158 |
| Sichuan | 0.215 | 0.342 | 0.413 | 0.366 | 0.356 | 0.629 | 0.715 | 0.628 |
| Guizhou | 0.659 | 0.542 | 0.366 | 0.564 | 0.631 | 0.397 | 0.459 | 0.547 |
| Yunnan | 0.403 | 0.458 | 0.521 | 0.415 | 0.39 | 0.365 | 0.362 | 0.252 |
| Tibet | 0.445 | 0.357 | 0.257 | 0.417 | 0.401 | 0.325 | 0.339 | 0.256 |
| Shaanxi | 0.43 | 0.386 | 0.32 | 0.59 | 0.496 | 0.483 | 0.72 | 0.392 |
| Gansu | 0.355 | 0.411 | 0.62 | 0.477 | 0.753 | 0.622 | 0.55 | 0.386 |
| Qinghai | 0.633 | 0.575 | 0.936 | 0.49 | 0.614 | 0.464 | 0.298 | 0.213 |
| Ningxia | 0.524 | 0.714 | 0.603 | 0.478 | 0.325 | 0.418 | 0.355 | 0.227 |
| Xinjiang | 0.491 | 0.56 | 0.702 | 0.642 | 0.475 | 0.316 | 0.337 | 0.213 |

eastern regions' swift development had generated a radiative effect, catalyzing progress in the central region. Based on their vast natural resources and labor force, these regions saw significant advancement, with Anhui Province (0.930) attaining the highest integration score. Hubei (0.722) and Hunan (0.708) were close behind, emerging as critical regions for urban–rural integration development. The 2022 national policies supporting western regions are expected to encourage further development. Yunnan (0.941), the Guangxi Zhuang Autonomous Region (0.935), and the Tibet Autonomous Region (0.933) had elevated integration indices, highlighting these western areas' significant potential, signifying a new stage in Chinese urban–rural interactions.

Geographical proximity's influence was apparent across many provinces. Zhejiang, a frontrunner in urban–rural integration with a development index of 0.662, attained economic prosperity and social advancement while impacting nearby Jiangsu (0.648) and Anhui (0.758), enhancing their rural–urban integration. This effect transcends the eastern regions. Sichuan Province, a pivotal western-area contributor, was a robust driving force with an integration level of 0.678,

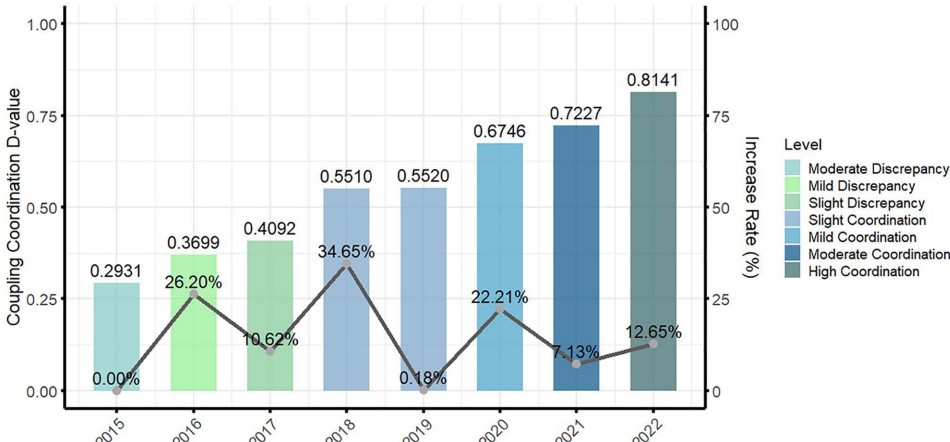

**Fig 3. The temporal changes in urban–rural integration development in China (2015–2022).**

promoting advancement in adjacent Yunnan (0.677) and Guizhou (0.722). Thus, interrelated urban–rural development dynamics appear nationwide (Figs 4–6).

## Primary driving factors of urban–rural integration development in China

We examined several main factors (infrastructure, economic development, public services, social security, and the ecological environments) that are expected to have a significant effect on urban–rural integration in China. We found that among these, the most significant drivers were broadband Internet availability, per capita disposable income, education spending, unemployment insurance coverage, and green ecological coverage. Within the infrastructure category ($X_1$–$X_3$), the urban–rural broadband Internet access comparison coefficient ($X_2$) had the highest q value of 0.6498. In economic development ($X_4$–$X_6$), the urban–rural per capita disposable income comparison coefficient ($X_4$) had the highest q value of 0.5258. In public services ($X_7$–$X_9$), the coefficient for the urban–rural per capita education spending ($X_7$) was notable, with a q value of 0.4360. In social security ($X_{10}$–$X_{12}$), the comparative coefficient for urban–rural unemployment insurance coverage ($X_{12}$) had the highest q value of 0.5198. Finally, in the ecological environment ($X_{13}$–$X_{15}$), the urban–rural green coverage comparison coefficient ($X_{13}$) had the highest q value at 0.4213. These findings highlight the unique contributions of each factor in urban–rural integration. These results are shown in Fig 7.

## Discussion

### Key findings

**Analysis of temporal evolution characteristics.** The overall development of urban-rural integration in China is on the rise, but the growth rate is uneven, displaying a wave-like evolution pattern of "rapid growth-slow development-fluctuating rise." The reasons behind this phenomenon are analyzed as follows: First, the release of policy dividends is the core driving force behind the rapid development of urban-rural integration. The country has introduced a series of top-level design documents such as the "Opinions on Establishing and Improving the System and Policy Framework for Urban-Rural Integration," with breakthroughs in land system reforms (such as allowing rural collective commercial construction land to enter the market) and household registration system reforms (such as fully lifting urban-rural household registration restrictions) [6,50]. These reforms have broken long-standing institutional barriers that constrained the free flow of resources. This loosening of regulations has significantly stimulated market vitality. By 2018, the rural land transfer rate reached 36.79%, an increase of 21.18% compared to before the policy was implemented [51]. Meanwhile, the

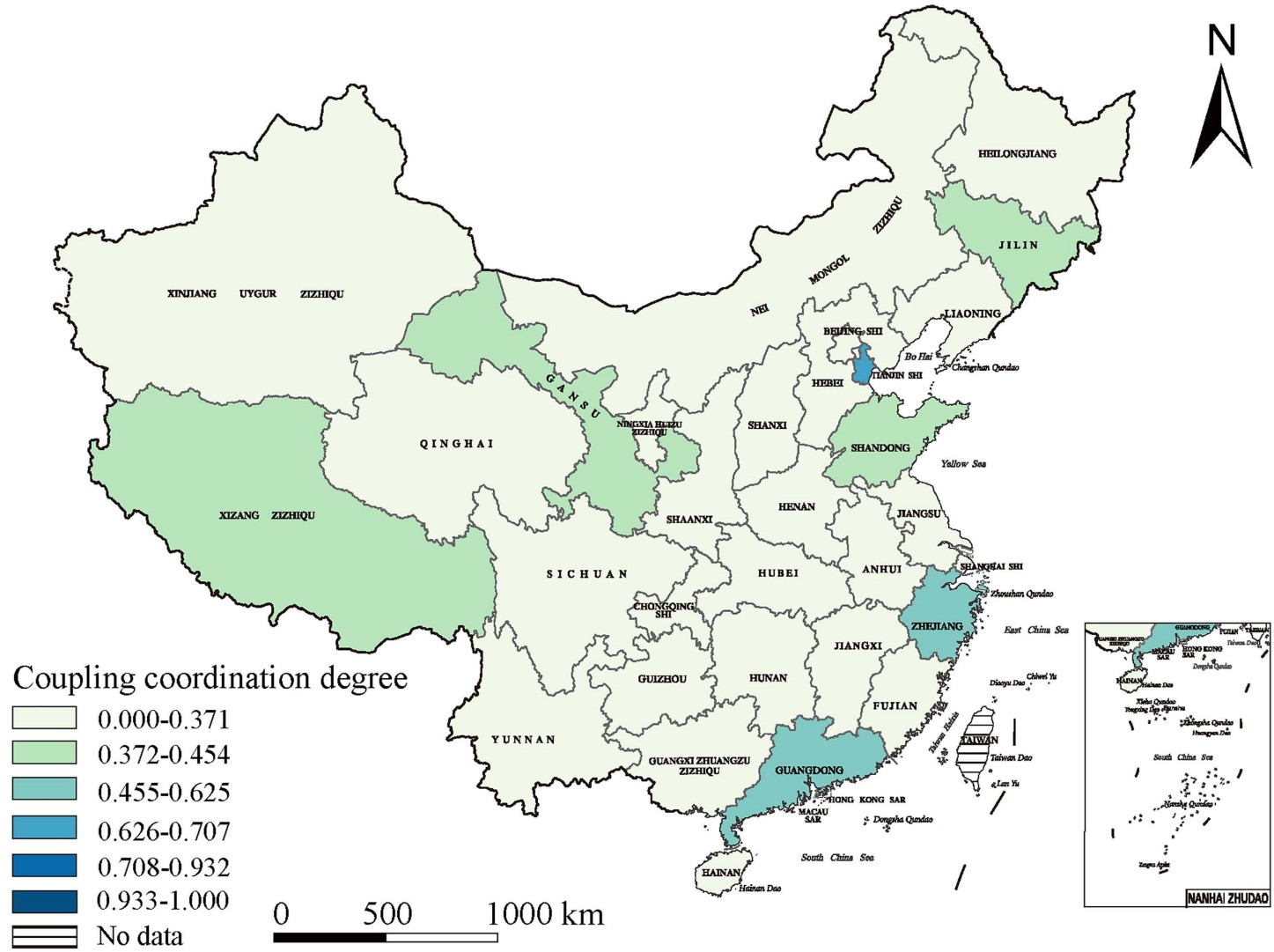

**Fig 4. The spatial distribution of urban–rural integration development in 2015.** (The base map of this figure uses public domain data from Natural Earth (available at https://www.naturalearthdata.com/), and the map was created using ArcGIS).

number of migrant workers returning to rural areas to start businesses reached 7.8 million, a 62.5% increase compared to before the policy was implemented, effectively activating the endogenous driving force of rural development [52,53]. On this basis, the policy has focused on increasing fiscal transfer payments to address gaps in rural public services. In the field of education, in 2018, the per-student educational expenditure for rural middle school students was 13,912 yuan, an increase of 41.2% compared to before the policy was implemented [54]. Meanwhile, the rate of rural participation in the new rural cooperative medical insurance reached 99.45%, an increase of 0.56% compared to before the policy was implemented, narrowing the gap in basic public services between urban and rural areas and providing solid livelihood guarantees for urban-rural integration [55]. Furthermore, the policy has focused on promoting the deep integration of urban and rural industries. By cultivating modern agricultural industrial parks, building characteristic towns, and other platforms, it has guided urban capital and technology to flow into rural areas, integrating them with rural resource endowments and ecological advantages. By 2018, the added value of rural secondary and tertiary industries accounted

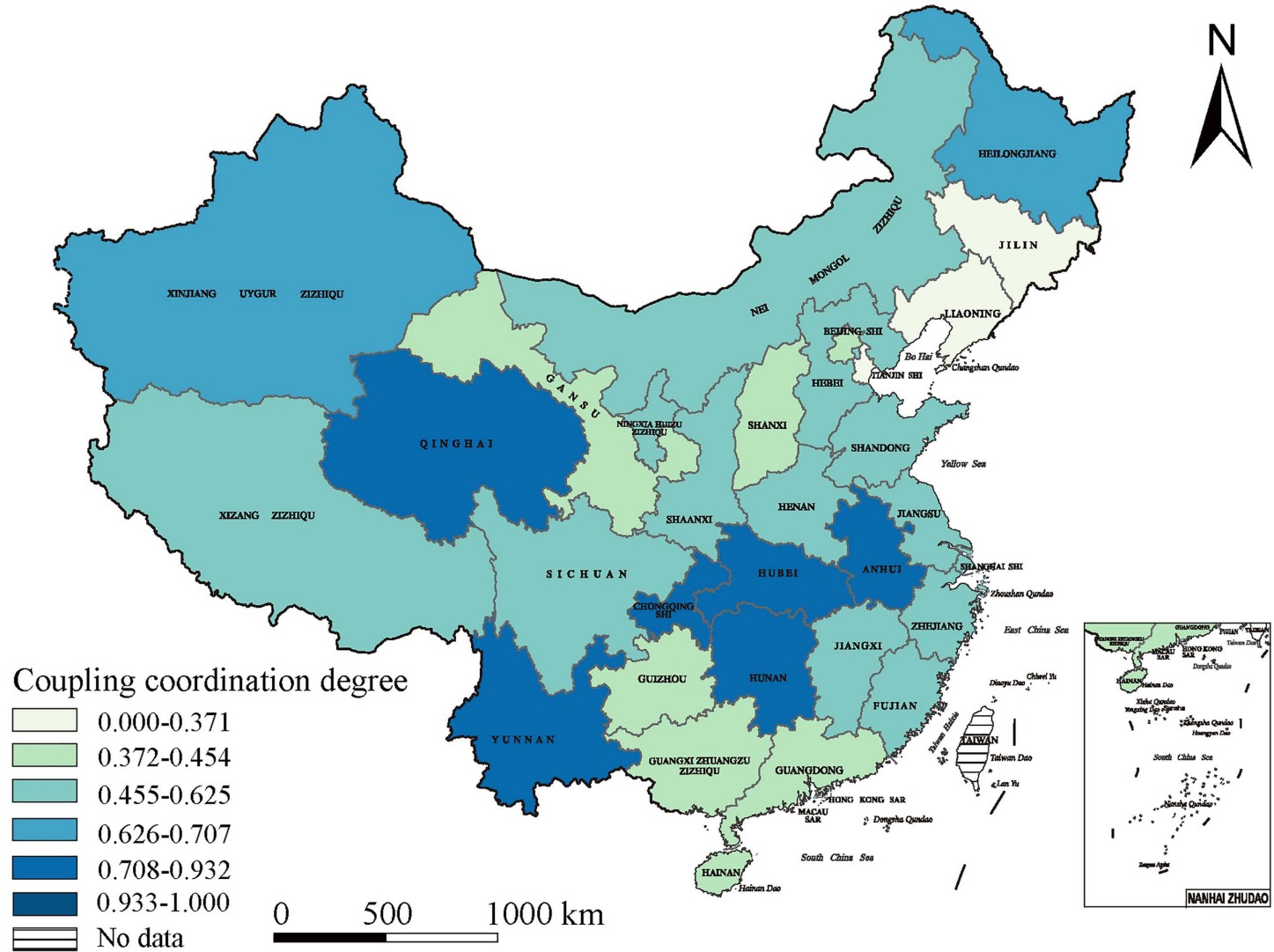

**Fig 5. The spatial distribution of urban–rural integration development in 2019.** (The base map of this figure uses public domain data from Natural Earth (available at https://www.naturalearthdata.com/), and the map was created using ArcGIS).

for 92.8% of GDP, an increase of 2.3% compared to before the policy was implemented, gradually forming a new pattern of linked industrial development between urban and rural areas [56]. Compared to existing studies, this research not only validates the role of policies proposed by Ma in promoting urban-rural integration [6], but also reveals that the policies mainly promote rapid urban-rural integration through a four-dimensional mechanism of "institutional breaking-resource activation-service balancing-industry linkage."

However, after entering 2019, the outbreak of the COVID-19 pandemic became a key obstacle to the slow development of urban-rural integration. In order to prevent the spread of the virus, the government implemented strict transportation controls and restrictions on interregional movement, which temporarily hindered the normal flow of production factors between urban and rural areas. By 2020, the number of migrant workers leaving for work was only 169.59 million, a decrease of 2.67% compared to the previous year [57]. Meanwhile, the agricultural product logistics system was inefficient, leading to a disconnection between regional production and sales of fresh agricultural products. In some major

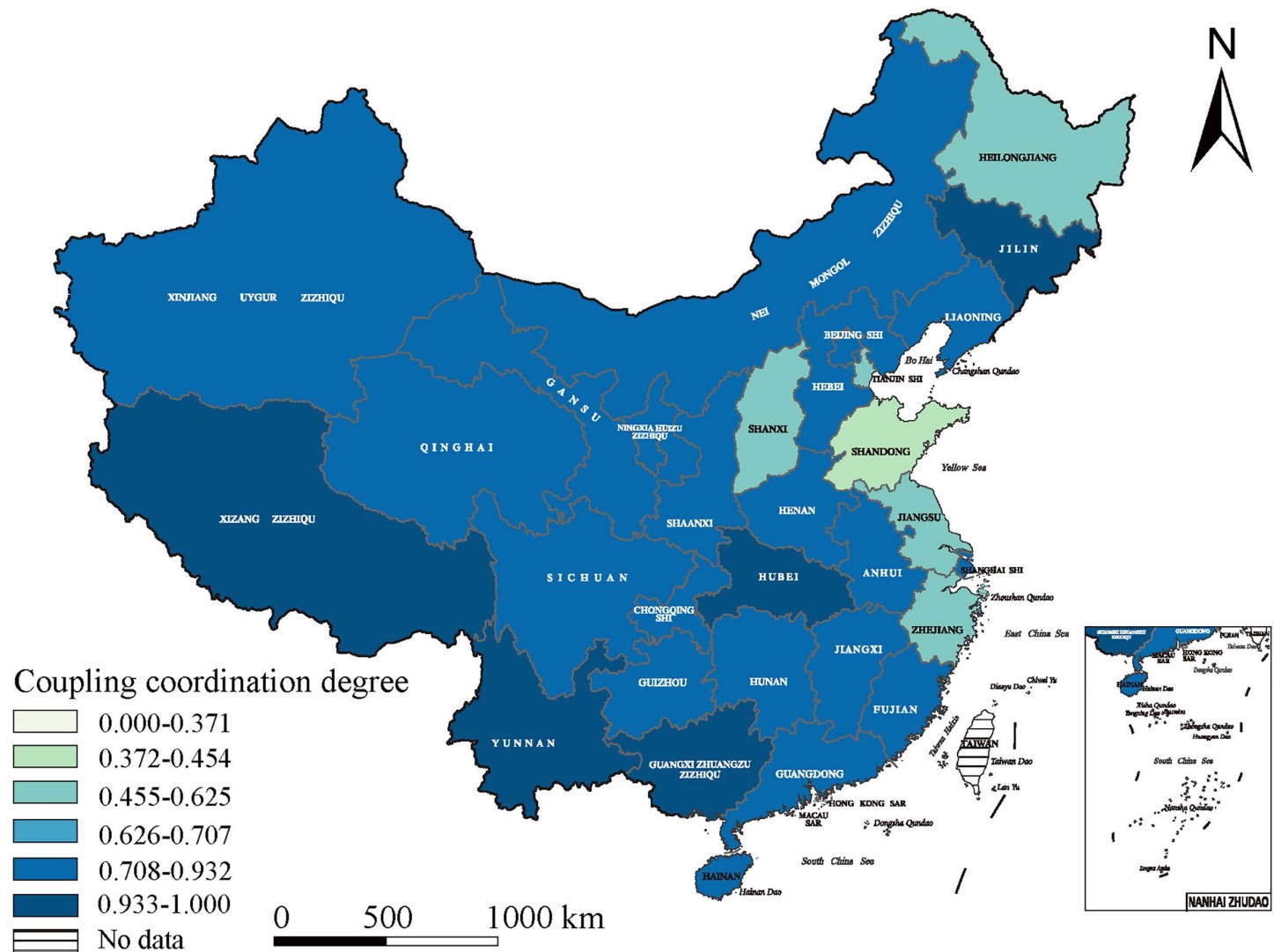

**Fig 6. The spatial distribution of urban–rural integration development in 2022.** (The base map of this figure uses public domain data from Natural Earth (available at https://www.naturalearthdata.com/), and the map was created using ArcGIS).

agricultural production counties, the unsold rate exceeded 25%, significantly weakening the urban-rural economic cycle and causing a temporary decline in rural household incomes [58]. Against this backdrop, emerging rural businesses relying on labor mobility and scene-based consumption were particularly hard-hit. By 2020, rural tourism revenue amounted to 600 billion yuan, a 66.9% decrease compared to the previous year [59]. A large number of small businesses and individual operators, dependent on cultural and tourism services, faced existential crises [60]. Compared to existing studies, this research not only identifies the pandemic as an exogenous shock, as emphasized by Li [57], but also further reveals that its impact mainly slowed down the urban-rural integration process through three core aspects: "disrupting the flow of resources—weakening the urban-rural economic cycle—impacting the rural industrial ecosystem."

Economic structural adjustment is the key driving force behind the "fluctuating upward" trend in urban-rural integration development [25]. Alongside market demand contraction and the widespread adoption of digital technologies, traditional heavy industries and the energy sector are generally facing declining efficiency and shrinking employment, which has,

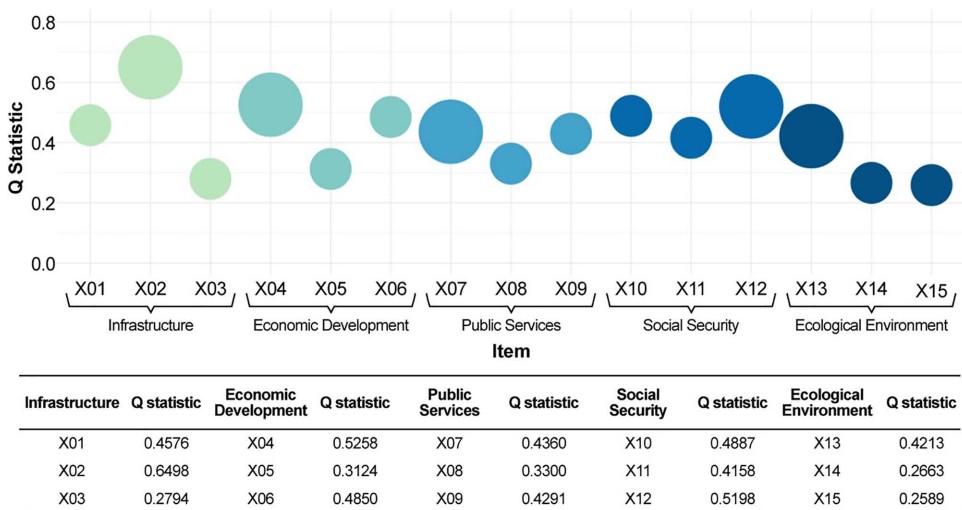

**Fig 7. The main driving factors of urban–rural integration in China (2015–2022).**

| Infrastructure | Q statistic | Economic Development | Q statistic | Public Services | Q statistic | Social Security | Q statistic | Ecological Environment | Q statistic |
|---|---|---|---|---|---|---|---|---|---|
| X01 | 0.4576 | X04 | 0.5258 | X07 | 0.4360 | X10 | 0.4887 | X13 | 0.4213 |
| X02 | 0.6498 | X05 | 0.3124 | X08 | 0.3300 | X11 | 0.4158 | X14 | 0.2663 |
| X03 | 0.2794 | X06 | 0.4850 | X09 | 0.4291 | X12 | 0.5198 | X15 | 0.2589 |

to some extent, exacerbated the income gap between urban and rural areas, creating phase-specific pressures on the integration process. At the same time, emerging business models such as leisure agriculture and rural tourism have flourished, not only driving the continued expansion of e-commerce for agricultural products but also accelerating the influx of social capital into new rural sectors [61]. These new industries effectively absorb the production factors released by the transformation of traditional sectors, creating a large number of new employment opportunities in rural areas, significantly expanding farmers' income channels, and injecting sustained and powerful endogenous momentum into urban-rural integration [62]. In this transformation process, there is a clear dynamic tug-of-war between the decline of traditional industries and the rise of emerging industries: the short-term pain caused by the adjustment of traditional sectors intertwines with the long-term benefits brought by the expansion of the new economy [63]. However, as the growth potential and driving effects of the new driving forces continue to strengthen, their upward momentum generally surpasses the downward pressure brought by the decline of traditional sectors, thus driving the overall "fluctuating upward" development trend of national urban-rural integration [18]. Compared to existing research, this paper not only verifies the profound impact of economic structural adjustments on urban-rural integration development, as proposed by Xu [25], but also deepens the understanding of the structural causes behind the "fluctuating upward" trend in urban-rural integration from the perspective of the transition between old and new driving forces.

**Analysis of spatial distribution characteristics.** The urban-rural integration development in China shows distinct spatial heterogeneity, forming a diverse development pattern where the eastern coastal region leads, the central region rises rapidly, and the western region has great potential. This spatial structure is closely related to the basic conditions and development paths of different regions. The eastern coastal regions, benefiting from superior geographical locations, have actively developed an open economy, gradually building a solid economic foundation, attracting large amounts of capital and high-end industries, and effectively promoting the early development of urban-rural integration through industrial gradient transfer mechanisms. Taking Zhejiang Province as an example, this province is located on the southeast coast and is home to the Ningbo-Zhoushan Port, the largest cargo throughput port in the world [64]. Its unique geographical advantages provide a solid foundation for the development of an open economy. Leveraging this condition, Zhejiang has vigorously developed port industries and cross-border e-commerce, gradually building a strong economic base. By 2022, Zhejiang's per capita GDP reached 118,496 yuan, ranking fourth among all provinces and regions in China [65].

On this basis, the province has attracted a large amount of capital and high-end industries by improving infrastructure and continuously optimizing the business environment. However, as the costs of land, labor, and other factors in the central cities continue to rise, many high-value-added industries such as export processing and smart manufacturing have gradually spread to counties and rural areas with larger development spaces and lower overall costs [66]. This process has significantly driven industrial upgrading and expanded employment opportunities in rural areas, effectively narrowing the urban-rural development gap. By 2022, the urban-rural income ratio in Zhejiang had dropped to 1.90, significantly lower than the national average, making it one of the provinces with the most significant achievements in urban-rural coordinated development [67]. As pointed out by Wang's research, the economic development level plays a key role in urban-rural integration [64]. However, this study further emphasizes that urban-rural integration not only depends on the overall economic level but is also closely related to the restructuring of industrial space, actively promoting urban-rural integration towards a higher level of development.

The central region, benefiting from its abundant labor resources, has effectively aligned with the market demands of the eastern coastal regions. An increasing number of labor-intensive industries such as textiles and apparel, electronics, and home appliance manufacturing have chosen to relocate to the central region, where factors like land and labor costs have a comparative advantage. In this process, large amounts of capital, technology, and management expertise have continuously flowed into the central region through direct investment, industrial transfer, and regional cooperation, not only driving the transformation and upgrading of local traditional industries but also accelerating the agglomeration of emerging industries [6]. It is particularly noteworthy that these industrial transfers have not only remained at the city level but have shown significant "county-oriented" characteristics. Taking Henan Province as an example, the province has actively taken on industrial transfer from the Yangtze River Delta region by building modern industrial clusters, creating several specialized industrial clusters at the county level [68]. For example, the textile and apparel industrial cluster in Taikang County, Zhoukou City, not only attracted well-known enterprises from the eastern region but also extended the industrial chain to rural areas through a "leading enterprise + rural workshop" model, directly providing employment and income growth for nearly 100,000 rural laborers [69]. The smart sensor industrial cluster formed in Nanyang City, in addition to setting up core research and manufacturing links around the city, has also dispersed supporting links such as packaging, logistics, and basic component production to surrounding towns, promoting the deep integration and collaborative development of urban and rural industrial chains [70]. As pointed out by Ma's research, industrial transfer has injected strong momentum into urban-rural integration [6]. However, this study further emphasizes that the deep embedding of industries at the county and rural levels not only promotes the reallocation of production factors between urban and rural areas but also significantly drives urban-rural integration towards higher quality development through job creation and industrial chain collaboration.

In contrast, the western region faces more prominent urban-rural dual structure issues due to its inland location and complex ecological environment. However, with the continuous support of national strategies such as the Western Development Program, poverty alleviation, and rural revitalization, the region has made significant progress in breaking through the bottleneck of urban-rural integration [71]. Taking Sichuan Province as an example, its "characteristic agriculture + rural tourism" dual-driven model in the Qinba Mountains and Liangshan Yi Autonomous Prefecture has not only activated rural industries but has also become an important mechanism for connecting urban and rural areas and promoting integration [72]. On one hand, by developing specialty industries such as highland agriculture and ecological breeding, and leveraging ethnic culture and ecological resources to promote rural tourism, rural areas have gradually established high-value-added industrial systems, strengthening their connections with urban consumer markets [73]. On the other hand, the opening of e-commerce channels has not only achieved the smooth flow of both "agricultural products going up" and "industrial products going down," but has also profoundly transformed the traditional urban-rural exchange model, promoting the more fair and efficient flow of resources and elements between urban and rural areas [74]. These measures have not only promoted income growth for farmers and diversified rural economies but have also fundamentally reshaped

urban-rural relations. Rural areas are no longer a simple resource exporter or a passive recipient of development but are gradually growing into development subjects that complement urban functions and share value [8]. In this process, characteristic industries have become the link for urban-rural industrial collaboration, e-commerce platforms have become bridges for connecting urban and rural markets, and policy guidance has provided a solid guarantee for institutional integration between urban and rural areas. As pointed out by Wang's research, policy support plays a key role in driving urban-rural integration [71]. However, this study further emphasizes that the integration of urban and rural areas in the western region not only depends on external resource inputs but, more fundamentally, on the construction of endogenous driving mechanisms. Only by fully unleashing the local development potential of rural areas can urban-rural integration move towards a more sustainable direction.

**Analysis of driving factors.** The development of urban-rural integration in China is influenced by multiple factors, including infrastructure, economic development, public services, social security, and the ecological environment. Among these, broadband internet access, per capita disposable income, education expenditure, unemployment insurance coverage, and green coverage rate have played key roles in promoting this development.

Specifically, broadband internet access is the cornerstone of urban-rural information resource sharing, directly impacting the speed of information transmission, knowledge acquisition, and cultural exchange. However, due to under-developed infrastructure, the broadband access rate in rural areas is generally lower than in urban areas, leading to a significant digital divide [75]. Therefore, reducing the gap in broadband access between urban and rural areas is crucial for achieving information sharing and further promoting urban-rural integration.

Per capita disposable income, as an important indicator of economic development, directly reflects the consumption capacity of urban and rural residents. However, due to the simple industrial structure and backward production technology in rural areas, per capita disposable income is generally lower than in urban areas, which continues to widen the income gap between urban and rural areas [76]. This income disparity not only lowers the consumption capacity of rural residents but also hinders the overall economic development of rural areas. Therefore, narrowing the gap in per capita disposable income between urban and rural areas is of paramount importance for improving the level of economic development and promoting urban-rural integration.

Education expenditure is a key means of improving the quality of human capital in urban and rural areas. However, rural areas currently face a shortage of educational investment, primarily due to the low income levels of rural families, which makes it difficult for them to bear high educational costs [77]. This disparity in educational resources not only exacerbates the inequality in educational opportunities between urban and rural residents but also severely hinders the overall improvement of human capital quality in rural areas. Therefore, reducing the gap in education expenditure between urban and rural areas is crucial for enhancing human capital quality and promoting urban-rural integration.

Unemployment insurance coverage is an important indicator of the completeness of the social security system in urban and rural areas. However, due to inadequate institutional design, the unemployment insurance coverage rate in rural areas is generally lower than in urban areas [78]. This disparity not only leaves rural residents inadequately protected against the risk of unemployment but also limits the economic development of rural areas. Therefore, narrowing the gap in unemployment insurance coverage between urban and rural residents is crucial for improving the social security system and promoting urban-rural integration.

Green coverage rate is an important indicator of regional ecological environmental quality. However, due to economic advantages and policy support, urban areas generally have a higher green coverage rate than rural areas [79]. This disparity not only results in unequal access to green ecological benefits for urban and rural residents but also hinders the improvement of the ecological environment and quality of life in rural areas. Therefore, narrowing the gap in green coverage rate between urban and rural areas is crucial for improving regional ecological environmental quality and promoting urban-rural integration.

## Innovation

The potential innovation of this research lies in abandoning the traditional paradigm of viewing urban and rural areas as two separate units and separately addressing their respective development issues. Instead, it employs a systems theory approach, viewing them as a closely connected and mutually interactive organic whole. By deeply investigating the development level, core characteristics, and influencing factors of urban-rural integration, this research enhances the understanding of the internal mechanisms of urban-rural integration and provides a solid empirical foundation for formulating strategies for urban-rural integration development. On this basis, the study further expands the traditional three-dimensional analytical framework of economy, society, and ecology, innovatively incorporating two additional dimensions: infrastructure and public services, thus constructing a five-dimensional analytical framework encompassing economy, society, ecology, infrastructure, and public services. This upgrade not only significantly enhances the comprehensiveness of urban-rural integration development level assessment but also provides policymakers with a more detailed reference framework, helping to implement more scientific, rational, and efficient policy measures.

## Limitations and future research directions

However, this study faces several limitations during the research process, which need to be addressed in future studies. On one hand, although efforts were made to comprehensively reveal the current status of urban-rural integration development, the constructed indicator system does not fully cover important dimensions such as population mobility, land use, and cultural integration, which to some extent limits the understanding of the complex mechanisms of urban-rural integration. On the other hand, due to the reliance on statistical yearbook data, which inherently have a time lag, the data used in this study only extend up to 2022, making it difficult to capture the latest development dynamics and affecting the timeliness of the research findings. To address these issues, future research can be improved in the following two aspects: At the indicator construction level, the urban-rural integration evaluation system should be continuously refined, incorporating micro-level case studies to better capture dimensions such as population mobility, land use, and cultural integration, thereby enhancing the systematic nature of the indicators. At the data acquisition level, it is recommended to introduce big data or remote sensing data to compensate for the limitations of traditional statistical sources in terms of timeliness and coverage, enabling higher-frequency and more precise measurement of the urban-rural integration process.

## Conclusions and recommendations

### Conclusions

Based on panel data from 31 provinces, autonomous regions, and municipalities directly under the central government in China from 2015 to 2022, this study systematically examines the level of urban-rural integration development, spatiotemporal evolution characteristics, and driving mechanisms in China, using methods such as the entropy method, coupling coordination degree model, and geographic detector. The main research conclusions are as follows:

(1) From the temporal perspective, the overall level of urban-rural integration development in China has been rising, but the growth process has shown significant fluctuations, with prominent phase characteristics. Specifically, the evolution trajectory follows a pattern of "rapid growth—slow development—fluctuating rise."

(2) From the spatial distribution perspective, there are significant regional differences in the development of urban-rural integration in China, presenting a multi-dimensional development pattern of "eastern regions developing first—central regions rising rapidly—western regions having huge potential."

(3) From the influencing factors perspective, urban-rural integration development in China is jointly affected by five dimensions: infrastructure, economic development, public services, social security, and the ecological environment. Among these, key factors driving integration development include internet broadband access rate, per capita disposable

income of rural residents, education expenditure level, unemployment insurance coverage rate, and urban-rural greening coverage rate.

## Recommendations

In light of the development characteristics of urban-rural integration in different stages in China, it is necessary to adopt targeted policy measures. During the rapid growth stage, the fundamental contradiction faced by urban-rural integration is the disconnect between traditional institutional frameworks and development needs. At this point, policies should focus on institutional breakthroughs and infrastructure construction to inject strong initial momentum into the integration process. Specifically, efforts should be made to vigorously promote rural land system reforms and the market-oriented allocation of factors, eliminating institutional barriers that hinder resource flow. At the same time, the construction of county-level industrial parks and digital infrastructure should be accelerated, actively attracting industrial spillovers from central cities. Systematic vocational skills training should be introduced to enhance the rural labor force's ability to migrate and find employment, laying a solid foundation for urban-rural integration. When the development stage shifts to slower growth, the extensive model of relying solely on factor input is no longer sustainable, and urban-rural integration urgently needs to transition from scale expansion to quality improvement. During this phase, policy focus should shift to fostering endogenous development momentum. On one hand, efforts should be made to promote the equalization of public services such as education and healthcare, reducing the social gap between urban and rural areas and creating a favorable environment for high-quality development. On the other hand, policies should actively cultivate emerging sectors such as rural tourism and e-commerce, encouraging rural industrial diversification and achieving a critical shift from "external support" to "self-sustaining growth." In the fluctuation and rising stage, the main challenge of urban-rural integration lies in insufficient system stability and weak risk resistance. Policies must focus on building risk buffering mechanisms and stabilizing development expectations, with two key aspects: First, establishing a dynamic monitoring and early warning system for urban-rural integration to promptly identify socio-economic risks, and innovatively using fiscal and financial tools for counter-cyclical regulation. For instance, rural infrastructure construction projects can be initiated at appropriate times to stabilize livelihoods during periods of increased employment pressure. Second, maintaining policy continuity and stability, strengthening government credibility, and providing market players with a predictable institutional environment to fundamentally enhance the sustainability of urban-rural integration.

Faced with the development characteristics of urban-rural integration in different regions of China, differentiated development strategies should be adopted. In the eastern coastal regions, efforts should focus on promoting the aggregation of high-end factors and breakthrough innovations, with an emphasis on developing smart agriculture, digital rural areas, and future communities. At the same time, traditional manufacturing industries should be guided to move in an orderly manner to the central and western regions, and key institutional barriers, such as land and household registration, should be overcome to provide replicable and scalable practical models for urban-rural integration across the country. In the central regions, there is a need to strengthen industrial undertaking and hub functions, accelerate the improvement of transportation logistics infrastructure and business environments, actively undertake industrial transfers from the east, cultivate county-level characteristic industrial clusters, explore the construction of "remote industrial zones" with eastern provinces, deepen cross-provincial industrial cooperation and talent exchange, and enhance regional collaborative development capabilities. In the western regions, central government fiscal support should play a leading role in speeding up the improvement of infrastructure and public service gaps. At the same time, leveraging ecological resource advantages, the focus should be on cultivating green industries such as ecological agriculture, clean energy, and cultural tourism, and creating a distinctive and sustainable integration development path.

In light of the various factors influencing urban-rural integration in China, it is necessary to focus on key influencing factors to achieve breakthroughs. In terms of improving internet broadband access, the "Broadband Rural" initiative should

be continuously promoted, along with subsidy policies for service fees, with the goal of achieving full coverage of gigabit optical networks in administrative villages by 2025, significantly improving the penetration rate of 100Mbps broadband in rural households. At the same time, digital service centers at the county level should be established, strengthening technical training and application guidance to ensure that broadband networks not only provide good connectivity but also deliver tangible benefits. In terms of increasing rural residents' per capita disposable income, the "One Township, One Product" income enhancement plan should be implemented, with each township focusing on cultivating 1–2 deep processing projects for characteristic agricultural products. Additionally, a rural e-commerce direct-to-market system should be established, setting up standardized service centers for agricultural products at the county and township levels, integrating packaging, testing, logistics, and other elements, and providing one-stop service support. In optimizing educational expenditure, the "County-Managed, School-Employed" teacher management reform should be deepened to ensure the benefits of rural teachers. At the same time, a pairing mechanism between urban and rural schools should be established, promoting the extension of high-quality educational resources to rural areas, thus facilitating the balanced development of education between urban and rural areas. In expanding the coverage of unemployment insurance, migrant workers and workers in new employment forms should be fully included in the security system, with a combined model of "government subsidies + enterprise contributions + voluntary individual participation" for insurance enrollment. At the same time, through the nationwide unified social security service platform, the seamless transfer and continuation of unemployment insurance relationships across regions should be realized, effectively enhancing the ability to cope with unemployment risks. In terms of improving urban-rural greening coverage, the rural greening project should be steadily promoted with annual growth targets. Meanwhile, a systematic approach to urban-rural ecological restoration should be carried out to continuously improve living conditions in both urban and rural areas, enhancing the quality of the ecological environment.

## Supporting information

**S1 File. The dataset after collection and calculation.**
(XLSX)

## Author contributions

**Conceptualization:** Wei Wang.

**Data curation:** Hui Liu.

**Formal analysis:** Hui Liu.

**Funding acquisition:** Wei Wang.

**Methodology:** Hui Liu.

**Software:** Hui Liu.

**Supervision:** Hui Liu.

**Visualization:** Hui Liu.

**Writing – original draft:** Hui Liu.

**Writing – review & editing:** Wei Wang.

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
