## [Decision Letter · Decision Letter 0]

1 Aug 2025

Dear Dr. Wang,

Thank you for submitting your manuscript to PLOS ONE. After careful consideration, we feel that it has merit but does not fully meet PLOS ONE’s publication criteria as it currently stands. Therefore, we invite you to submit a revised version of the manuscript that addresses the points raised during the review process.

While the reviewers commend the relevance of your work on urban-rural integration and its potential contributions to regional planning, major revisions are required to address significant methodological and analytical concerns before further consideration. Key issues include:

(1) Insufficient justification for defining and applying "comparative coefficient" indicators, including ambiguous ideal values and inconsistent directional attributes;

(2) Lack of theoretical depth in framing urban-rural theories and inadequate literature engagement, particularly regarding research gaps;

(3) Unexplained data volatility (e.g., abrupt drops in provincial integration indices) and methodological inconsistencies in weighting sub-systems;

(4) Superficial discussion of findings, policy implications, and driving mechanisms (e.g., broadband internet’s role); and

(5) Low-resolution figures and non-intuitive data presentation (Table 3).

Please also address all specific critiques from Reviewers (e.g., abstract clarity, sample details, indicator references, spatial statistics).

We look forward to receiving your revised manuscript.

Kind regards,

Najmeh Mozaffaree Pour, Ph.D. 

Guest Editor

PLOS ONE

“This research was funded by National Social Science Late Grant Program, grant number 22FGLB007 and Project of Natural Science Foundation of Hunan Province, grant number 2023JJ30313.”

4. We note that Figures 2, 4, 5 and 6 in your submission contain [map/satellite] images which may be copyrighted. All PLOS content is published under the Creative Commons Attribution License (CC BY 4.0), which means that the manuscript, images, and Supporting Information files will be freely available online, and any third party is permitted to access, download, copy, distribute, and use these materials in any way, even commercially, with proper attribution. For these reasons, we cannot publish previously copyrighted maps or satellite images created using proprietary data, such as Google software (Google Maps, Street View, and Earth). For more information, see our copyright guidelines: http://journals.plos.org/plosone/s/licenses-and-copyright.

1. You may seek permission from the original copyright holder of Figures 2, 4, 5 and 6 to publish the content specifically under the CC BY 4.0 license. 

5. We note that there is identifying data in the Supporting Information file < Supplementary Information.xlsx>. Due to the inclusion of these potentially identifying data, we have removed this file from your file inventory. Prior to sharing human research participant data, authors should consult with an ethics committee to ensure data are shared in accordance with participant consent and all applicable local laws.

-Location data

Please remove or anonymize all personal information, ensure that the data shared are in accordance with participant consent, and re-upload a fully anonymized data set. Please note that spreadsheet columns with personal information must be removed and not hidden as all hidden columns will appear in the published file.

Reviewers' comments:

Reviewer's Responses to Questions

**Comments to the Author**

1. Is the manuscript technically sound, and do the data support the conclusions?

Reviewer #1: Yes

Reviewer #2: Yes

Reviewer #3: Yes

2. Has the statistical analysis been performed appropriately and rigorously?

Reviewer #1: I Don't Know

Reviewer #2: Yes

Reviewer #3: Yes

3. Have the authors made all data underlying the findings in their manuscript fully available?

Reviewer #1: Yes

Reviewer #2: Yes

Reviewer #3: Yes

4. Is the manuscript presented in an intelligible fashion and written in standard English?

Reviewer #1: Yes

Reviewer #2: Yes

Reviewer #3: Yes

Reviewer #1: The article is well structured and well written. It complies adequately with the recommended scientific structure and presents a satisfactory explanation in terms of methodology and its replicability.

The theme is pertinent and necessary to the field of regional planning and territorial planning, being essential to deny the theme as a duality (urban vs. rural), but rather as an important continuum between parts in territory. Recognising that urban studies and planning need to include a better understanding of the rural and its interactions is essential at the current moment in the field of knowledge. The conception of these socio-spatial relations as a system is pertinent and adequate. A good example of research applicable to other realities within the same scope. The global comparative analysis is well developed and appropriate.

It would be interesting if the article provided more specific information about the findings, such as concrete examples of the regions compared and their specificities, thus making it clearer how public policies could be implemented to improve the performance evaluated.

Reviewer #2: Overall Comments:

This study provides an in-depth analysis of the spatiotemporal evolution and driving factors of urban-rural integration in China, utilizing quantitative methods such as the entropy method, coupling coordination degree model, and geographical detector, based on data from 31 provinces between 2015 and 2022. However, the study has shortcomings in the rigor of its indicator system construction, the depth of result interpretation, and the critical engagement with existing literature in the discussion section. Critical issues include explaining data volatility and justifying the definition and rationale of “comparative coefficient” indicators. Given the impact of these issues on the research conclusions, I recommend a major revision.

Specific Comments:

1.Introduction. The introduction effectively sets the stage by discussing the evolution of urban-rural relations and the policy context of urban-rural integration in China. However, the description of existing research gaps could be more specific. For instance, beyond stating the lack of “comprehensive national-level study” and content concentration on “traditional dimensions,” it could further explain how these gaps hinder a comprehensive understanding of the complexity of urban-rural integration and how this study’s “five-dimensional analytical framework” addresses these deficiencies. Emphasize the unique theoretical and practical value of this research, for example, how it deepens the understanding of urban-rural factor flow and balanced public resource allocation based on existing literature.Consider citing more official documents or authoritative interpretations of China’s new-era urban-rural development strategies, such as in-depth research on the “Rural Revitalization Strategy”, It is recommended that the references include citations of literature, such as mentioned by Ma et al., (2025) (https://doi.org/10.1371/journal.pone.0313125), Yu et al., (2025) (https://doi.org/10.1038/s41598-025-08492-3), Yang et al., (2021) (https://doi.org/10.1016/j.habitatint.2021.102420)

2.Literature review. The literature review traces the conceptual origins of urban-rural integration and outlines the dimensions, methods, and scope of existing research, identifying gaps. However, the theoretical framework discussion could be more in-depth. For example, when mentioning “Urban Bias Theory,” “Rural Bias Theory,” “Urban-Rural Dual Structure Theory,” and “Urban-Rural Integrated Development Theory,” briefly elaborate on their core tenets and contributions to understanding urban-rural relations, explaining how this study extends or engages with them. Furthermore, when pointing out the absence of “infrastructure” and “social security” dimensions in existing research, elaborate on why these two dimensions are crucial for understanding urban-rural integration and connect them with other existing theories or empirical findings. For the “Urban-Rural Dual Structure Theory,” cite its classic work to strengthen the theoretical foundation. Such as Lewis, W. A. (1954). Economic development with unlimited supplies of labour. The Manchester School, 22(2), 139-191.

3.Methods. The methods section provides a detailed description of the study area, data sources, indicator descriptions, and models.

(1)Indicator Descriptions: This is the section most in need of major revision. The definition and application of “comparative coefficients” are central. All indicators in the study use “comparative coefficients,” which are ratios of urban to rural values. Firstly, a clear definition of how these “comparative coefficients” are specifically calculated is needed. For example, is “Urban–rural road length comparison coefficient”“Urban road length / Rural road length” or the inverse?Secondly, and more importantly, there needs to be sufficient justification for using these ratio indicators to measure urban-rural integration. If the ideal state of urban-rural integration implies a reduction in disparities between urban and rural areas, then an ideal value for a ratio indicator should approach 1. However, in the indicator descriptions, for instance, “Urban–rural greening coverage comparison coefficient” is marked with “�” (negative attribute), while “Urban–rural pension insurance coverage comparison coefficient” is marked with “�” (positive attribute). These indicate their direction of impact but do not explain their ideal value or how they reflect “integration.” Please clearly explain how these ratios are transformed into measures of urban-rural integration and ensure logical consistency across all indicators in terms of their attribute (positive or negative) and ideal integration state. For example, if a larger urban/rural ratio means greater disparity, then such indicators should be inversely treated when calculating the composite index. Rationale for indicator selection: While the dimensions are expanded, the representativeness of some indicators might need further consideration. For example, the “Urban–rural per capita cultural and recreational expenditure comparison coefficient” might exhibit high volatility, questioning its stability in describing long-term trends of urban-rural integration.

(2)Data Processing: Using linear interpolation for a small amount of missing data is acceptable, but its limitations should be acknowledged, and a brief discussion of potential implications for the results, even if minor, should be included.

(3)Coupling Coordination Degree Model:The study assumes that “the five sub-systems hold equal significance; hence, we allocate a value of 1/5 for each.” While common, this assumption requires more robust justification. The entropy weighting method has already calculated weights for individual indicators, and the composite index scores of these sub-systems (U1-U5) are likely derived based on these entropy weights. If U1-U5 are already weighted results from the entropy method, then applying equal weights (1/5) again for the β values in calculating T might introduce a logical inconsistency or double-weighting. Please clarify the calculation process of U1-U5 and whether the β values should be combined with entropy weights, or provide sufficient reasons for adopting equal weights at the sub-system level.

4.Results.

(1)The title "Results" in section 4 should be changed to " Results Analysis ".

(2)Temporal Evolution: In Table 3, some provinces show significant fluctuations and even substantial declines in urban-rural integration (e.g., Beijing from 0.893 in 2018 to 0.439 in 2020; Zhejiang from 0.735 in 2017 to 0.561 in 2018). Such drastic drops for an index measuring “integrated development” are highly noteworthy. The authors need to explain the reasons for these anomalous fluctuations. Are they data issues? Methodological limitations? Or do they genuinely reflect a real regression? This section requires a more detailed explanation beyond simply generalizing it as a “fluctuating rise.”

(3)Spatial Evolution: The spatial analysis describes regional disparities and proximity effects. To enhance the rigor of the analysis, consider incorporating spatial statistics (e.g., Moran’s I) to quantify and verify the observed spatial autocorrelation or clustering patterns, rather than solely relying on descriptive statements.

(4)Driving Factors: Figure 7 clearly displays the q-values for the main driving factors. In the textual description, beyond listing these factors, a deeper analysis of their relative importance (e.g., broadband internet access having the highest q-value, indicating its strongest explanatory power) and a brief discussion of the underlying mechanisms would be beneficial.

5.Discussion. The discussion section interprets the results and attempts to compare them with existing literature.

(1)Depth and Criticality: The discussion should delve deeper into the underlying reasons and mechanisms behind the research findings. For example, regarding the impact of “policy support,” “COVID-19 pandemic,” and “economic structural changes” on urban-rural integration, it should more specifically explain how they led to the observed “rapid growth—slow development—fluctuating rise” pattern and how they explain significant declines in certain provinces. Simply stating that “this viewpoint aligns with XX’s research” is insufficient; further discussion is needed on how this study’s findings “deepen” or “complement” existing research, or in what aspects they offer “new insights.”

(2) Geographical Proximity Effect: While the geographical proximity effect is discussed, its specific transmission mechanisms could be further explored. Is it through the flow of talent and capital, or through information and technology diffusion? Is it a spillover effect or a competitive effect?

(3)Policy Implications: Policy recommendations are a significant contribution of this study, but they should be more closely linked to the specific research findings. For instance, given that broadband internet access is a main driving factor, the policy recommendations should more prominently highlight how to specifically enhance broadband penetration and utilization efficiency in rural areas, rather than just vaguely mentioning “upgrading information infrastructure.” More actionable recommendations, such as differentiated policies targeting various development stages and regional characteristics, could be proposed.

6.Conclusions.The conclusion section summarizes the main findings, potential innovations, and research limitations. (1) Main Findings: While comprehensive, the summary of main findings could be more concise and succinct, reiterating the most core discoveries. (2) Innovation: The study elaborates well on its innovations, especially “abandoning the traditional binary paradigm” and “expanding the five-dimensional analytical framework.”This part is well-written and these innovations should be highlighted earlier and more confidently in the introduction and literature review. In the conclusion, these innovations can be more succinctly summarized as to how they “fill existing research gaps” and “provide a more scientific, rational, and efficient policy reference.”(3) Limitations and Future Research: The discussion of data lag and insufficient coverage is candid. For future research directions, beyond “refining and optimizing the evaluation index system” and “actively collecting and utilizing the latest statistical data,” more specific data sources and research methods could be proposed. For example, using big data or remote sensing data to compensate for the limitations of statistical yearbooks, or conducting micro-level case studies to delve into complex issues such as population mobility, land use, and cultural integration.

7.Grammar and Phrasing. The overall language of the study is fluent, but some long sentences could be further refined to improve readability. Conduct another round of professional English proofreading to ensure precision and academic rigor.

8.Figure and Table Presentation.

(1)All embedded images in the study, including Figures 1 to 7 (e.g., coupling coordination mechanism diagram, element level charts, regional heterogeneity charts, etc.), generally suffer from low resolution. The image edges appear blurry, and text and details are not clear enough. This significantly compromises the professionalism and readability of the study, and does not meet the publication standards for high-quality academic journals. I strongly recommend that the authors submit images in higher resolution formats for the final submission. TIFF format (typically requiring at least 300 DPI) or EPS format (vector graphics) are recommended.

(2)Currently, Table 3 (Index of urban–rural integration development across China regions from 2015 to 2022) presents a large amount of data in tabular format. While providing detailed numerical values, it makes it difficult to intuitively discern the temporal trends of each province and to perform cross-sectional comparisons among different provinces.It is strongly recommended to convert the data in Table 3 into more visual chart formats.

Reviewer #3: 1. In Abstract: The presented abstract appears to be comprehensive and coherent, but some expressions require further explanation or simplification. Some sentences are long and somewhat complex. Similar concepts, such as development trends and influencing factors, are repeated several times. Additionally, it would be beneficial to provide more details about the volume and type of data and the studied samples in the research methodology section.

2. In section 1, the problem statement is sufficiently general and indicates that the topic is important. However, to enhance clarity and achieve a deeper understanding of the issue, more detailed explanations regarding the challenges, gaps, and consequences of the problem are necessary. This approach helps the reader or reviewer gain a better understanding of the needs and main objectives of the research.

3. From line 89 onwards, it is necessary to refer to relevant research in the areas you have mentioned, and their findings should also be presented.

4. End of research literature, line 98, given that many studies have been conducted on urban-rural integration, particularly concerning China, what are the research gaps and innovative aspects of this study?

5. In Data Sources, line 114, please specify the Time Period of the received information.

6. In section Indicator Descriptions, more and more relevant references need to be used.

7. Table 1, Attribute Weight, Please explain the negative and positive values of the weights. In some cases, they do not seem logical.

8. In section 4.3., it is necessary to examine the obtained numbers by mentioning the desired model.

9. In the discussion section, it is necessary to discuss the research findings more clearly, and it is also recommended to use more references.

**Do you want your identity to be public for this peer review?** For information about this choice, including consent withdrawal, please see our Privacy Policy

Reviewer #1: **Yes: ** Vânia Raquel Teles Loureiro

Reviewer #2: No

Reviewer #3: **Yes: ** Saeedeh Moayedfar, Associate Professor, Meybod University

---

## [Author Response · Author response to Decision Letter 1]

26 Oct 2025

For detailed information, please refer to the file "Response to Reviewers."

---

## [Decision Letter · Decision Letter 1]

6 Nov 2025

The spatiotemporal evolution and driving factors of urban–rural integration over the past eight years in China Evidence from 31 provinces

PONE-D-25-17463R1

Dear Dr. Wang,

We’re pleased to inform you that your manuscript has been judged scientifically suitable for publication and will be formally accepted for publication once it meets all outstanding technical requirements.

Kind regards,

Najmeh Mozaffaree Pour

Guest Editor

PLOS ONE

Additional Editor Comments (optional):

Reviewers' comments:

Reviewer's Responses to Questions

**Comments to the Author**

Reviewer #2: All comments have been addressed

Reviewer #3: All comments have been addressed

2. Is the manuscript technically sound, and do the data support the conclusions?

Reviewer #2: Yes

Reviewer #3: Yes

3. Has the statistical analysis been performed appropriately and rigorously?

Reviewer #2: Yes

Reviewer #3: Yes

4. Have the authors made all data underlying the findings in their manuscript fully available?

Reviewer #2: Yes

Reviewer #3: Yes

5. Is the manuscript presented in an intelligible fashion and written in standard English?

Reviewer #2: Yes

Reviewer #3: Yes

Reviewer #2: The authors have revised the manuscript following the reviewers' comments and suggestions. I have no further suggestions.

Reviewer #3: I have re-evaluated the manuscript and found that my previous concerns have been satisfactorily addressed. The authors have revised the manuscript in response to the reviewer’s comments, and the issues identified in the prior iteration appear to be resolved.

The study design, methods, and analyses are now clearly described, and the data presented support the conclusions drawn.

The manuscript is currently suitable for publication, and I have no further objections. I recommend accepting the manuscript, subject to any minor editorial adjustments the journal editor may require.

**Do you want your identity to be public for this peer review?** For information about this choice, including consent withdrawal, please see our Privacy Policy

Reviewer #2: No

Reviewer #3: No

---

## [Editor Report · Acceptance letter]

PONE-D-25-17463R1

PLOS ONE

Dear Dr. Wang,

I'm pleased to inform you that your manuscript has been deemed suitable for publication in PLOS ONE. Congratulations! Your manuscript is now being handed over to our production team.

Kind regards,

on behalf of

Dr. Najmeh Mozaffaree Pour

Guest Editor

PLOS ONE